# Comparing the Effects of AI-Assisted and Traditional Exercise on Physical Health Outcomes in Older Adults: A Systematic Review and Meta-Analysis

**DOI:** 10.3390/healthcare13232999

**Published:** 2025-11-21

**Authors:** Sijing Fan, Xin Tan, Hongyun Zheng, Yicong Cui, Xiaotong Du, Boqiao Huang, Jingzhan Ren, Xinming Ye, Wen Fang

**Affiliations:** 1School of Sports, Science and Engineering, East China University of Science and Technology, 130 Meilong Road, Xuhui District, Shanghai 200237, China; y30231582@mail.ecust.edu.cn (S.F.); y30241614@mail.ecust.edu.cn (H.Z.); tong20000529@gmail.com (X.D.); jkyjs@mail.ecust.edu.cn (X.Y.); 2College of Smart Materials and Future Energy, Fudan University, 220 Handan Road, Yangpu District, Shanghai 200433, China; xtan21@m.fudan.edu.cn; 3College of Engineering and Design, Hunan Normal University, 36 Lushan Road, Yuelu District, Changsha 410081, China; 4Division of Sports Science and Physical Education, Tsinghua University, No. 30 Shuangqing Road, Haidian District, Beijing 100084, China; cyc23@mails.tsinghua.edu.cn; 5China Japan Union Hospital, Jilin University, No. 2699 Qianjin Street, Changchun 130012, China; huangbq9922@mails.jlu.edu.cn; 6School of Business, East China University of Science and Technology, 130 Meilong Road, Xuhui District, Shanghai 200237, China; y30231329@mail.ecust.edu.cn

**Keywords:** artificial intelligence, network meta-analysis, older adults, physical function, cognitive function, quality of life

## Abstract

Objective: Exercise is widely recognized as an effective non-pharmacological intervention to maintain health in older adults. With advances in artificial intelligence (AI), AI-assisted exercise has emerged as a novel rehabilitation approach, yet its comparative effectiveness against traditional and software-assisted programs remains unclear. This study aimed to evaluate and rank the relative effectiveness of these interventions on multiple physical and psychological outcomes using a network meta-analysis (NMA). Methods: Following the PRISMA-NMA guidelines, we systematically searched PubMed, Embase, Cochrane Library, Web of Science, and Scopus up to June 2025. Eligible studies were randomized controlled trials (RCTs) involving adults ≥ 60 years comparing AI-assisted, software-assisted, and conventional upper/lower limb rehabilitation. Six outcomes were analyzed: gait, balance, range of motion (ROM), muscle strength, cognitive function, and quality of life (QOL). Stata 17.0 was used to conduct the NMA, calculating the standardized mean differences (SMDs) and SUCRA rankings, with assessments of heterogeneity and risk of bias. Results: Seventy RCTs with 808 participants were included. All active interventions outperformed the placebo. AI-assisted programs showed the strongest effects on gait (SMD = 1.33) and balance (SMD = 0.76), while software-assisted interventions ranked highest for ROM (SMD = 0.69) and QOL (SMD = 1.06). Both AI and software interventions improved cognition and muscle strength. Heterogeneity was low (I^2^ ≤ 38.5%). Subgroup analysis indicated that AI-based methods were superior to traditional rehabilitation, although differences among novel interventions were not statistically significant. Conclusions: AI-assisted exercise is highly effective for gait and balance, while software-assisted approaches excel in ROM and QOL. These interventions hold promise for community and home-based rehabilitation. Future studies should investigate integrated “AI + traditional” models and incorporate biomechanical and neurophysiological indicators to optimize personalized care.

## 1. Introduction

With the ongoing shifts in global demographics, population aging has emerged as one of the most critical public health and social challenges of the 21st century. According to the World Population Prospects 2022 Summary by the United Nations, by 2050, the global population aged 60 years and above will exceed 2.1 billion, accounting for 22% of the total population, with the fastest growth observed among the “oldest-old” group aged 80 and above. This structural transformation is particularly pronounced in Europe, North America, and East Asia. The global trend of aging is characterized by a large scale, rapid growth, and significant regional disparities, posing profound impacts on national healthcare systems, rehabilitation resource allocation, and the equity of health services.

As people age, they are increasingly confronted with multisystem functional decline and comorbid chronic diseases, including sarcopenia, cognitive impairment, cardiovascular diseases, Parkinson’s disease, Alzheimer’s disease, depression, and multimorbidity [1]. These health issues often manifest in the early stages as functional deterioration, such as balance disorders, gait instability, joint mobility limitations, muscle weakness, and reduced quality of life, which severely impair daily independence and healthy lifespan [2]. Among these, falls resulting from functional decline are one of the most common and harmful outcomes in older adults. According to the World Health Organization, approximately one-third of individuals aged 65 and older experience at least one fall each year, making falls a leading cause of disability, hospitalization, and mortality in this population. Maintaining the basic physiological and cognitive functions of older adults has become a crucial strategy for preventing severe health deterioration, with non-pharmacological interventions receiving increasing attention and application.

Effective exercise interventions are regarded as the core non-pharmacological strategy to counteract multidimensional functional decline in older adults. Studies have shown that comprehensive intervention programs combining balance training, aerobic exercise, and strength training can effectively delay physical deterioration, reduce fall risk, and decrease the probability of disability [3]. Traditional exercise-based rehabilitation has been widely proven to enhance cardiovascular fitness, muscle strength, and balance while promoting neural plasticity, making it a critical approach for improving physical and mental health in the elderly.

However, traditional rehabilitation models face multiple limitations: they heavily rely on face-to-face guidance from therapists, which makes it challenging to meet the long-term, individualized, and home-based training needs of the large aging population. Additionally, the uneven distribution of rehabilitation resources across regions prevents older adults in rural areas, low-income communities, or with limited mobility from accessing sustained and effective intervention support [4].

The rapid advancement of Artificial Intelligence (AI) has introduced new possibilities to the field of exercise rehabilitation and is considered a key driver of the transition toward precision rehabilitation. In recent years, AI has been widely applied in posture recognition, gait analysis, scenario detection, and decision support systems for training, forming a novel intervention model that integrates data-driven analysis with personalized feedback [5]. Specifically, AI, combined with wearable devices, cameras, deep learning algorithms, and remote platforms, enables high-frequency collection, real-time analysis, and personalized feedback on exercise data for older adults, significantly improving the control of training intensity and execution accuracy [6]. Moreover, AI-based systems often incorporate gamification elements, visual interfaces, and reward mechanisms, effectively enhancing older adults’ engagement and adherence to training, thus addressing the traditional rehabilitation bottleneck of “low participation and high dropout rates” [7]. In home-based settings, the integration of AI with smart terminals for remote rehabilitation has been shown to improve both physical and cognitive functions in older adults, with high levels of user satisfaction and technological acceptance [8]. Additionally, AI systems assist in risk prediction and rehabilitation pathway recommendations during exercise interventions, facilitating the development of intelligent, continuous, and cost-effective rehabilitation services [9].

In this study, we distinguished between two major forms of technology-mediated exercise interventions. AI-assisted interventions were defined as those incorporating artificial intelligence algorithms, such as deep learning, posture recognition, gait analysis, or real-time feedback mechanisms. In contrast, software-assisted interventions referred to approaches such as virtual reality systems, wearable devices, or gamified platforms that do not involve real-time AI algorithms. Although current studies have preliminarily demonstrated the potential of AI to improve gait, cognitive function, and quality of life, the diversity of intervention forms and the lack of consistent evaluation indicators have led to a shortage of systematic evidence, revealing its comprehensive effects and comparative advantages across multidimensional health outcomes. Particularly in the context of the coexistence of AI interventions, software-assisted training, and traditional upper/lower limb rehabilitation, the relative efficacy rankings remain unclear [10,11,12] Large-scale, multidimensional, and standardized research is needed to clarify the applicability boundaries of these interventions.

Therefore, this study followed the PRISMA-NMA guidelines and systematically included 70 randomized controlled trials (RCTs), aiming to compare the relative effectiveness of AI-assisted exercise, software-assisted interventions, and traditional upper and lower limb rehabilitation across six key health outcomes: Gait, Balance, Range of Motion (ROM), Muscle Strength, Mental Health, and Quality of Life (QOL). Using network meta-analysis, this research comprehensively explored the potential advantages and applicable scenarios of various interventions, providing evidence-based support for individualized rehabilitation in older adults and policy references for building scalable, efficient, and digitalized rehabilitation service systems in the future.

## 2. Methods

### 2.1. Study Design

This study was conducted and reported in accordance with the PRISMA-NMA (Preferred Reporting Items for Systematic Reviews and Meta-Analyses for Network Meta-Analyses) guidelines and the registration ID is CRD42025643308 [13]. A systematic review and network meta-analysis were employed to compare the relative effectiveness of AI-assisted exercise and traditional exercise interventions in improving multidimensional physical and psychological health outcomes among older adults.

### 2.2. Literature Search Strategy

We systematically searched five electronic databases: PubMed, Embase, Cochrane Library, Web of Science, and Scopus, with the last search conducted in June 2025. To improve specificity, the search strategy used detailed artificial intelligence-related terms such as “artificial intelligence”, “machine learning”, “deep learning”, “neural networks”, “gait analysis”, and “posture recognition”, combined with exercise- and outcome-related keywords (“exercise”, “older adults”, “gait”, “balance”, “muscle strength”, “range of motion”, “cognitive function”, “quality of life”). The isolated keyword “AI” was not used to avoid capturing unrelated terms (e.g., “Ai Chi,” “T’ai Chi”). Boolean operators (AND, OR) were applied to ensure comprehensive coverage of relevant studies. Specific search terms can be seen in Appendix A.

### 2.3. Inclusion and Exclusion Criteria

Studies were eligible for inclusion if they met the following criteria: (1) participants were aged 60 years or older; (2) interventions included AI-assisted exercise, software-assisted rehabilitation, or conventional upper/lower limb rehabilitation training; (3) study design was a randomized controlled trial (RCT); and (4) at least one relevant physical or psychological outcome was reported, including gait, balance, joint range of motion (ROM), muscle strength, cognitive function, or quality of life (QoL) [14]. Exclusion criteria were as follows: non-RCT designs, interventions not involving exercise or AI components, absence of extractable quantitative data, or conference abstracts and duplicate publications [15]. Only rehabilitation-focused exercise programs were eligible. General fitness, wellness, or purely preventive exercise programs without an explicit rehabilitation objective or protocolized therapeutic components were excluded. None of the included trials were general fitness/preventive programs; all targeted rehabilitation goals or outcomes. The rationale for the intervention classification of the included studies can be seen in Appendix A.

### 2.4. Data Extraction and Quality Assessment

Two independent reviewers screened and extracted data, including author, publication year, sample size, intervention type, intervention duration, outcome measures, and assessment tools. Disagreements were resolved by a third reviewer. The methodological quality of included studies was assessed using the Cochrane Risk of Bias tool, which evaluates domains such as random sequence generation, allocation concealment, blinding, data completeness, and reporting bias. All the data extracted from the articles can be seen in Appendix A.

### 2.5. Statistical Analysis

All statistical analyses were conducted using Stata version 17.0 (StataCorp., College Station, TX, USA) [16]. A random-effects model for network meta-analysis was applied to integrate both direct and indirect comparisons across intervention types. Standardized Mean Differences (SMDs) with 95% confidence intervals (CIs) were used as effect size metrics [17]. Network plots illustrated the structure of treatment comparisons, while forest plots summarized the estimated effects. SUCRA (Surface Under the Cumulative Ranking Curve) values were calculated to rank the relative effectiveness of each intervention [18]. All the results from the Stata analyses can be seen in Appendix A.

### 2.6. Heterogeneity and Publication Bias Assessment

Between-study heterogeneity was assessed using the I^2^ and τ^2^ statistics, with I^2^ > 50% considered indicative of moderate to high heterogeneity. Funnel plots were used for preliminary assessment of publication bias [19]. If sufficient studies were available, Egger’s test was also conducted to statistically evaluate bias, ensuring the robustness and reliability of the findings [20].

### 2.7. Quality Assessment

Two authors (ZHY and CYC) independently assessed the methodological quality of the included studies using the Revised Cochrane Risk of Bias Tool for Randomized Trials (RoB2). In the event of discrepancies, a third reviewer (DXT) was consulted to resolve the differences and reach a consensus. The risk of bias assessment encompassed seven key domains: (1) random sequence generation (selection bias); (2) allocation concealment (selection bias); (3) blinding of participants and personnel (performance bias); (4) blinding of outcome assessment (detection bias); (5) incomplete outcome data (attrition bias); (6) selective reporting (reporting bias); and (7) other sources of bias.

## 3. Results

A total of 535 records were identified through database searches, including PubMed (n = 107), Embase (n = 43), Scopus (n = 37), Web of Science (n = 63), and the Cochrane Library (n = 285). After removing 192 duplicate records, 343 studies remained for title and abstract screening. Following this, 203 full-text articles were assessed for eligibility. Based on the predefined inclusion and exclusion criteria, 20 randomized controlled trials (RCTs) involving 808 participants were finally included in the network meta-analysis. The primary reasons for exclusion were ineligible study design (n = 28), inappropriate outcome measures (n = 34), non-conforming intervention types (n = 50), non-target population (n = 36), and unsuitable control group settings (n = 33). The study selection process is presented in Figure 1. The detailed results of the risk of bias assessment are illustrated in Figure 2 and Appendix A.

### 3.1. Gait Performance

A total of 18 randomized controlled trials involving 567 participants were included to evaluate gait performance. The main outcome indicators for gait were assessed through network meta-analysis, as shown in Figure 3A and Appendix A. The commonly used gait assessment tools included the 6 min walk test (n = 4, accounting for 35.71%), the 10 m walking test (n = 5, 28.57%), and the Tinetti Gait Scale (n = 2, 14.29%). The network meta-analysis revealed that all active interventions showed a positive effect direction compared with conventional rehabilitation (Placebo), as presented in Figure 4A and Appendix A. According to the SUCRA-based ranking, the results were as follows: (1) AI-assisted intervention showed the highest effect size (SMD = 1.33, 95% CI: 0.78–1.88, statistically significant); (2) software-based intervention showed a significant effect (SMD = 1.08, 95% CI: 0.55–1.60); and (3) single lower extremity rehabilitation also demonstrated a significant effect (SMD = 0.78, 95% CI: 0.34–1.21), as shown in Figure 5A and Appendix A. AI-assisted interventions exhibited medium-to-large effect sizes when compared with Placebo, software-assisted interventions, and lower extremity rehabilitation. However, the differences between AI-assisted interventions and the other two active treatments were not statistically significant, as the 95% confidence intervals crossed zero. The overall heterogeneity was moderate (I^2^ = 63.5%, τ^2^ = 0.245).

A two-layer subgroup analysis strategy was adopted in this study. First, comparisons were made between novel interventions (AI-assisted, software-assisted, and lower limb rehabilitation) and traditional rehabilitation. The results showed that the novel interventions had a significantly superior intervention effect (SMD = 1.050, 95% CI: 0.536–1.563), though with substantial heterogeneity (I^2^ = 75.30%). Second, comparisons were conducted within the novel intervention group to examine differences among AI-assisted, software-assisted, and lower extremity rehabilitation interventions. The results indicated a moderate but significant difference (SMD = 0.395, 95% CI: 0.155–0.636), with no observed heterogeneity (I^2^ = 0%), suggesting that the efficacy of the three novel interventions was relatively stable. However, no absolute superiority of AI was detected in direct comparisons. The between-group difference was statistically significant (Qb = 5.12, *p* = 0.024), further indicating that while AI-assisted interventions have a clear advantage over traditional methods, further optimization is still needed to enhance their superiority within novel intervention strategies.

No significant inconsistency was detected across the network based on the node-splitting approach (*p* > 0.05), indicating good agreement between direct and indirect evidence. SUCRA rankings remained stable across sensitivity analyses, and the exclusion of high-risk studies did not alter the pooled effects, supporting the robustness of the findings.

### 3.2. Balance Ability

A total of 14 randomized controlled trials involving 430 participants were included to evaluate balance ability. The primary outcome indicators related to balance were assessed through network meta-analysis, as illustrated in Figure 3B. The balance assessment tools utilized across the studies included the Berg Balance Scale (n = 5, accounting for 28.57%), the Timed Up and Go Test (n = 4, 35.71%), and the Tinetti Balance Scale (n = 2, 14.29%). The network meta-analysis results showed that all active interventions demonstrated positive effects compared with conventional rehabilitation (Placebo), as presented in Figure 4B. According to the SUCRA rankings, the comparative effect sizes were as follows: AI-assisted intervention yielded the highest effect size (SMD = 0.76, 95% CI: 0.63–2.16), followed by software-based intervention (SMD = 0.52, 95% CI: −0.06–1.11), assisted upper limb movement (SMD = 0.39, 95% CI: −0.15–0.93), and single lower extremity rehabilitation (SMD = 0.29, 95% CI: −1.02–1.60), as shown in Figure 5B. Among these comparisons, the AI-assisted intervention demonstrated a statistically significant advantage in multiple pairwise comparisons, particularly when compared to single lower extremity rehabilitation, with a moderate improvement in balance outcomes (SMD = 0.47, 95% CI: 0.00–0.94). The heterogeneity analysis indicated a high level of consistency across studies, with no significant between-study variability detected (I^2^ = 0.00%, τ^2^ = 0.0000, Q = 7.36, *p* = 0.8828).

Consistency tests using both global and local approaches confirmed no significant network inconsistency (*p* > 0.05). SUCRA probability rankings for balance outcomes were consistent across all model assumptions. Sensitivity analysis excluding smaller-sample studies yielded comparable results, further supporting the stability of the network estimates.

### 3.3. Range of Motion (ROM)

A total of 9 randomized controlled trials involving 328 participants were included to assess joint range of motion (ROM). The primary outcome data related to ROM were analyzed through network meta-analysis, as illustrated in Figure 3C. The assessment tools used for ROM included overall Range of Motion evaluations in 3 studies (42.86%), Passive ROM in 2 studies (28.57%), and Active ROM in 2 studies (28.57%). The network meta-analysis results showed that all active interventions demonstrated positive effect directions compared with conventional rehabilitation (Placebo), as presented in Figure 4C. According to the SUCRA rankings, the effect sizes were as follows: software-based interventions had the highest effect (SMD = 0.69, 95% CI: 0.13–1.25), followed by assisted upper limb movement (SMD = 0.59, 95% CI: 0.00–1.17), AI-assisted intervention (SMD = 0.42, 95% CI: −0.18–1.01), and single lower extremity rehabilitation (SMD = 0.19, 95% CI: −0.54–0.92), as shown in Figure 5C. No statistically significant differences were observed in the direct comparisons between interventions. Specifically, AI-assisted interventions did not demonstrate clear advantages over software-assisted rehabilitation (SMD = −0.27, 95% CI: −1.00–0.45) or lower limb rehabilitation (SMD = 0.23, 95% CI: −0.39–0.85). The heterogeneity analysis showed no significant variability across studies (I^2^ = 0.00%, τ^2^ = 0.0000, Q = 6.90, *p* = 0.548), indicating a high level of consistency in the results.

Node-splitting analysis revealed no inconsistency between direct and indirect evidence (*p* > 0.05). The SUCRA-based ranking pattern remained stable after sensitivity testing, and no major shifts were observed when excluding studies with higher bias risk, confirming result robustness.

### 3.4. Muscle Strength

A total of 9 randomized controlled trials involving 286 participants were included to evaluate muscle strength outcomes. The primary outcome data were synthesized through network meta-analysis, as shown in Figure 3D. The assessment tools included grip strength measurements in 4 studies (44%) and isometric force tests in 3 studies (33%). The network meta-analysis results demonstrated that all active interventions produced positive effects compared with conventional rehabilitation (Placebo), as illustrated in Figure 4D. According to the SUCRA rankings, AI-assisted interventions showed the highest effect size (SMD = 0.61, 95% CI: 0.16–1.07). When compared with assisted upper limb exercise, AI-assisted interventions demonstrated a consistent advantage across 3 studies (SMD = 0.26, 95% CI: −0.12–0.65). Similarly, compared to lower limb wearable devices, AI-assisted interventions also showed a stronger potential effect in improving muscle strength (SMD = 0.38, 95% CI: 0.13–1.25). Compared with software-assisted interventions, AI-assisted interventions exhibited a positive trend (SMD = 0.15, 95% CI: −0.68–0.99), though this difference did not reach statistical significance, as presented in Figure 5D. The consistency of the included studies was acceptable, with low heterogeneity (I^2^ = 8.81%, τ^2^ = 0.0131).

Global inconsistency testing indicated a coherent network structure (*p* > 0.05). SUCRA analysis confirmed that AI-assisted interventions consistently ranked highest in improving muscle strength. Sensitivity analysis showed no meaningful change in ranking order, demonstrating good model reliability.

### 3.5. Cognitive Function

A total of 10 randomized controlled trials involving 270 participants were included to assess the effects of five different interventions on cognitive function, as illustrated in Figure 3E. Seven cognitive outcome measures were reported across the included studies, utilizing three primary assessment tools: the SF-36 Mental Component Summary (MCS) was used in 3 studies (42.9%), the Mini-Mental State Examination (MMSE) appeared in 2 studies (28.6%), and the SF-36 Social Functioning subscale was reported in 2 studies (28.6%). The network meta-analysis demonstrated that all active interventions showed positive effects compared with conventional treatment (Placebo), as shown in Figure 4E. According to the SUCRA rankings, the effect sizes were as follows: AI-assisted interventions had an SMD of 0.35 (95% CI: −0.36 to 1.06), software-based interventions demonstrated the highest effect (SMD = 1.07, 95% CI: 0.25–1.89), assisted upper limb movement showed an SMD of 0.18 (95% CI: −0.57 to 0.93), and single lower extremity rehabilitation had an SMD of 0.82 (95% CI: 0.40–1.24), as presented in Figure 5E. The overall pooled analysis indicated that the interventions significantly improved cognitive function, with a combined effect size of SMD = 0.472 (95% CI: 0.205–0.738, *p* = 0.0005), representing a small to moderate effect. The heterogeneity among studies was low (I^2^ = 15.01%, τ^2^ = 0.0277), indicating consistent results across the included trials.

The network model demonstrated good consistency (*p* > 0.05), with no local inconsistencies identified. SUCRA probability distributions remained stable across different priors, and sensitivity testing confirmed that the removal of small-sample trials did not alter the significance of the pooled effect.

### 3.6. Health-Related Quality of Life (HRQoL)

A total of 9 randomized controlled trials involving 334 participants were included to evaluate the impact of different interventions on health-related quality of life (HRQoL), as shown in Figure 3F. Various assessment tools were used across the studies, with the most commonly employed being the SF-36 or SF-12 Physical Component Summary (PCS), reported in 5 studies (55%), and the EQ-5D Visual Analogue Scale (VAS), used in 2 studies (22%). The network meta-analysis demonstrated that all active interventions showed positive effects in improving HRQoL compared with conventional treatment (Placebo), as presented in Figure 4F. According to the SUCRA rankings, software-based interventions achieved the highest effect size (SMD = 1.06, 95% CI: 0.41–1.72), followed closely by AI-assisted interventions (SMD = 1.05, 95% CI: 0.55–1.55). Assisted upper limb movement also showed a beneficial effect (SMD = 0.84, 95% CI: 0.25–1.43). In direct comparisons, software-based interventions were significantly superior to lower limb wearable device interventions (SMD = 0.57, 95% CI: 0.06–1.09), as illustrated in Figure 5F. Overall, the results indicated that all intervention modalities had a positive impact on improving quality of life in older adults, with statistically significant overall effects. The network heterogeneity was moderate (I^2^ = 38.54%, τ^2^ = 0.0815), suggesting some degree of variability across the included studies.

No evidence of global or local inconsistency was found (*p* > 0.05), and the SUCRA results maintained identical ranking patterns after sensitivity re-analysis. The model fit indices indicated good convergence and stability, confirming the reliability of the HRQoL findings.

## 4. Discussion

This study analyzed 70 intervention trials involving a total of 808 participants, with an average age of approximately 63.6 years old. It systematically compared the effects of five intervention modalities—AI-assisted, software-assisted, upper and lower limb rehabilitation, and conventional rehabilitation (Placebo)—across six key outcome indicators. Overall, the core finding is that all AI-assisted exercise interventions demonstrated superior effectiveness compared to conventional rehabilitation in improving gait, balance, joint range of motion, muscle strength, cognitive function, and quality of life. Among these, AI and software-assisted interventions frequently ranked in the top two positions according to the SUCRA probabilities, with the most significant improvements observed in gait (SMD = 1.33) and balance (SMD = 0.76). Our findings suggest that AI-assisted interventions are the most effective in enhancing key functional outcomes such as gait, balance, and quality of life in older adults, showing greater overall efficacy than both conventional and other digital rehabilitation methods. These results support the use of AI as a central component in building individualized and efficient geriatric rehabilitation systems.

The leading performance of AI and software-assisted interventions across multiple dimensions may be attributed to three core mechanisms of AI: (1) real-time precision feedback, where AI models capture various movement parameters such as gait cycles, postural stability, and reaction times, enabling dynamic evaluation and correction when compared to existing databases [21]; (2) adaptive training dosing, where AI algorithms adjust intensity based on real-time data from motion sensors, EMG, or EEG, achieving personalized rehabilitation [22]; (3) gamification and motivational designs, including score systems, visualized progress bars, and reward feedback, which enhance participation and adherence [23]. In contrast, traditional rehabilitation—although effective for targeting specific muscle groups—lacks cross-joint coordination, multi-sensory feedback, and simulated complex tasks, resulting in relatively lower engagement and comprehensive impact [24]. Notably, aside from gait (I^2^ = 63.5%), heterogeneity in other outcomes was low (I^2^ ≤ 38.5%), indicating robust and consistent findings. Further analysis showed that gait outcome differences among AI, software, and lower limb rehabilitation (SMD = 0.395, I^2^ = 0%) were relatively stable, with no absolute superiority of AI, suggesting that traditional lower-limb rehabilitation still holds specific benefits. Future research should explore combined modalities (e.g., “AI + lower limb wearable devices”) to maximize complementary strengths.

Prior studies show that motion-capture and body-sensing technologies can be feasibly deployed in institutional care settings and effectively enhance physical function and stability [25]. Additionally, mobile-based remote intervention platforms offer continuous training for home-dwelling older adults, achieving good adherence and intervention outcomes [26]. Our findings suggest that AI and software interventions are ideal for fall prevention at home, while community and care facilities can adopt low-cost sensing devices. For those with limited mobility, AI-driven remote interventions reduce resource demands and improve service accessibility, especially among high-risk populations [27]. Community-level deployments of low-cost motion-sensing devices can also address disparities in traditional rehabilitation services due to geographical and manpower constraints. Data collected by these devices can feed into deep learning algorithms to track gait cycles, correct movements in real-time, and optimize training based on multimodal monitoring, thereby enhancing overall training effectiveness [28]. Hence, while AI shows promise, traditional rehabilitation retains irreplaceable value.

For outcomes like joint range of motion and muscle strength, traditional upper/lower limb rehabilitation retains advantages, especially in terms of user acceptance among older adults. Active ROM and isometric strength training remain mainstream for enhancing localized muscle control, particularly in those with cognitive limitations or reduced fine motor skills [29]. While AI can collect kinematic and physiological signals (e.g., EMG, accelerometry) for real-time feedback, standardized applications for single-muscle or upper limb coordination remain underdeveloped [30]. Existing systems mainly focus on lower limb gait and balance, lacking multi-task upper limb interventions [31]. Thus, future approaches may incorporate “modular training” by embedding AI within therapist-led protocols, integrating ROM, isometric, and progressive resistance modules tracked by wearables to allow for data-driven feedback and dosage adjustment [32]. Incorporating neuro-feedback or movement-intent recognition could further boost AI’s ability to intervene in complex musculoskeletal functions [33].

AI also shows promising effects in improving cognitive function and quality of life. Wearables and remote digital platforms not only enhance physical performance but also improve cognition, emotional well-being, and tech acceptance [34]. These findings align with the “motor-cognition-psychological synergy” hypothesis: improved motor abilities activate neural circuits and plasticity, thereby enhancing cognitive states and subjective well-being [35]. This approach holds potential for early cognitive decline and psychological stress, especially in at-risk elderly populations. Future work should integrate objective neural indicators (e.g., fNIRS + EEG) to evaluate intervention mechanisms [36], and deep learning models (e.g., CNN-LSTM, Transformer + EEG) could jointly model behavior, physiology, and cognition. This would help clarify AI’s impact while reducing workforce burden and promoting cost-effective, scalable cognitive rehabilitation [37]. Applications may target homes, communities, and care institutions, integrated with fall prevention and real-world contexts. Despite these promising findings, several practical considerations should be acknowledged before large-scale implementation. The cost of AI-driven systems and wearable sensors may limit adoption in low-resource settings, and older adults with limited digital literacy may face barriers to use. Ensuring accessibility through simplified interfaces, government-subsidized equipment, and community-level training programs will be essential to promote equity

Future research should also incorporate biomechanical data, such as center of pressure (COP) and 3D gait analysis, to improve assessment sensitivity and precision. Studies show that 3D skeletal tracking using depth cameras can quantify joint angles, gait cycles, stride length, and coordination, supporting high-resolution functional evaluations and tailored therapy adjustments [38]. COP trajectory analysis is widely used to assess postural stability, especially among stroke patients or fall-prone elderly individuals. Key parameters (e.g., sway area, speed, amplitude) significantly change after intervention and reflect treatment efficacy [39]. Integrating COP with AI-learning models could enhance sensitivity to physiological changes, support therapy planning, and aid in outcome prediction and risk stratification. Future implementation should also consider integration into existing rehabilitation workflows, therapist training, and ethical governance regarding data privacy and algorithm transparency.

## 5. Strengths and Limitations

This study has several notable strengths. First, it is one of the first network meta-analyses to systematically compare the effects of multiple exercise interventions—including AI-assisted, software-assisted, and traditional upper and lower limb rehabilitation—on six core health outcomes in older adults: gait, balance, joint range of motion (ROM), muscle strength, cognitive function, and quality of life. It includes 70 randomized controlled trials involving a total of 808 participants with an average age of 63.6 years, providing a comprehensive and solid foundation for multidimensional outcome assessment. Second, by employing network meta-analysis along with SUCRA ranking and subgroup analysis, this study not only reveals overall intervention effectiveness but also explores subtle differences within emerging technologies, enhancing the clinical relevance of the findings. Third, except for the gait outcome, the heterogeneity across studies for all other outcomes was relatively low (I^2^ ≤ 38.5%), indicating high consistency and robustness of the results. Moreover, all outcome measures used in the included studies were based on standardized clinical scales, further supporting the reliability and generalizability of the conclusions.

Despite these strengths, several limitations should be acknowledged. First, although a substantial number of studies were included, some had small sample sizes, short follow-up periods, and moderate-to-high risk of methodological bias, which may affect the overall credibility of the evidence. Second, the classification of intervention types (e.g., AI-assisted, software-assisted, upper/lower limb rehabilitation) was based primarily on the information reported by the authors, which may have led to some ambiguity or overlap in categorization due to hybrid technologies. Third, although a comprehensive and systematic literature search was conducted, there remains a time lag between the search and publication of this review, which means that some newly published studies may not have been included, potentially impacting the completeness of the evidence. Finally, while this study emphasized multidimensional health outcomes, it did not account for variables such as intervention cost, accessibility, or adherence, which are crucial for evaluating the practical implementation of different rehabilitation strategies. Future research should incorporate health economics assessments to better inform policy decisions and intervention planning.

## 6. Conclusions

This study systematically included 70 randomized controlled trials involving 808 participants with an average age of approximately 63.6 years. Using a network meta-analysis framework (NMA), it compared the effects of AI-assisted, software-assisted, and traditional upper and lower limb rehabilitation on six major physical and cognitive outcomes in older adults. The results indicate that all active intervention strategies were more effective than conventional care. These findings suggest that AI systems may serve as a complementary and scalable tool in community and home-based rehabilitation to enhance accessibility and adherence. Future research should further explore hybrid multimodal strategies combining “AI + traditional rehabilitation” to improve precision, adaptability, and long-term functional outcomes.

## Figures and Tables

**Figure 1 healthcare-13-02999-f001:**
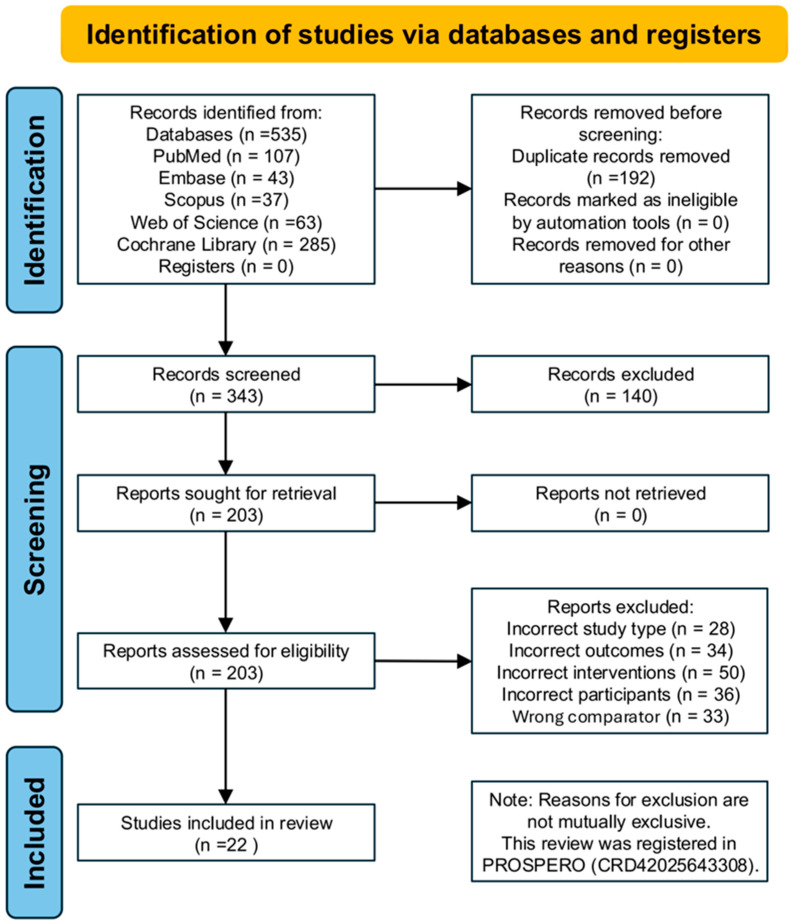
Study flow diagram.

**Figure 2 healthcare-13-02999-f002:**
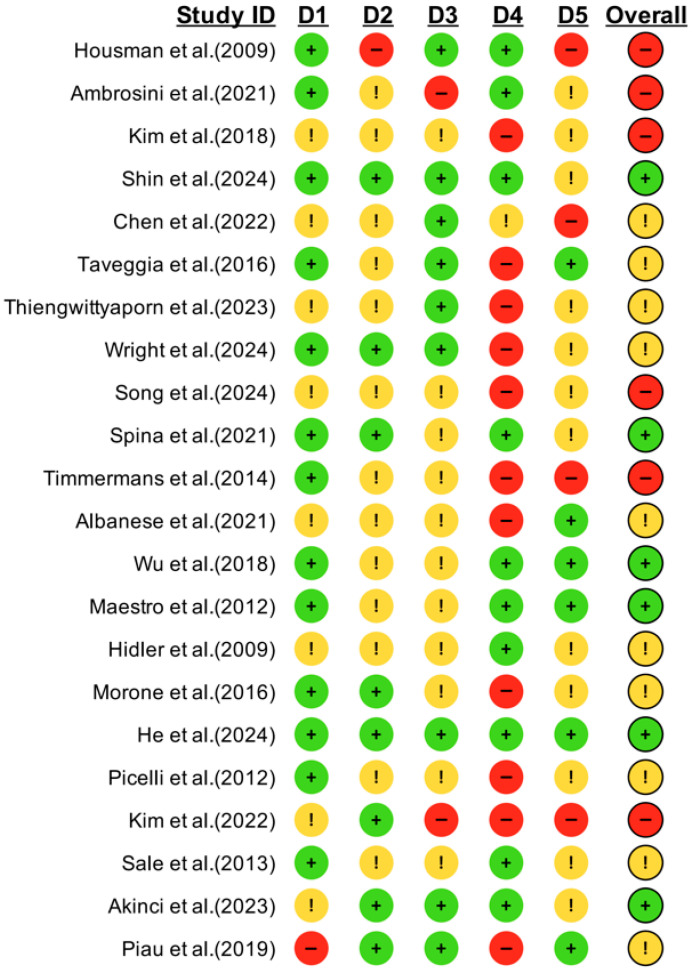
Risk of bias assessment for included randomized controlled trials (RCTs) using the Cochrane Risk of Bias tool. Each row represents an individual study, showing evaluations across six domains. **D1:** Random sequence generation. **D2:** Allocation concealment. **D3:** Blinding of participants and personnel. **D4:** Blinding of outcome assessment. **D5:** Incomplete outcome data. Overall: Overall risk of bias judgment.

**Figure 3 healthcare-13-02999-f003:**
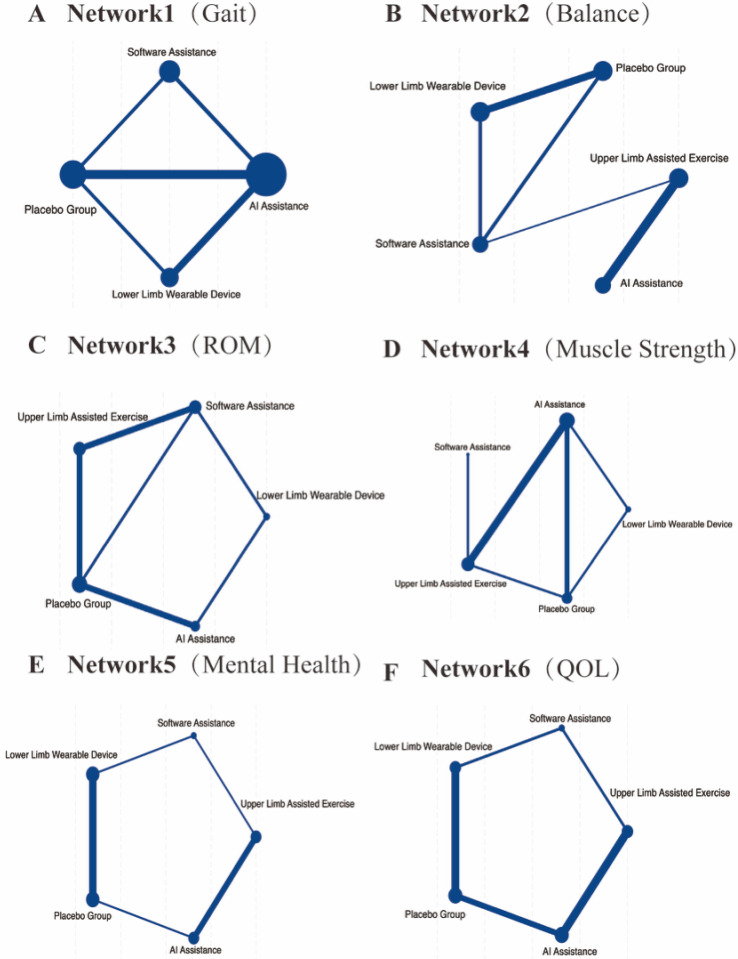
Network diagrams of included interventions for six health outcomes in older adults: (**A**) Gait Performance, (**B**) Balance Ability, (**C**) Joint Range of Motion (ROM), (**D**) Muscle Strength, (**E**) Mental Health, and (**F**) Quality of Life (QOL). Each node represents a type of intervention: AI Assistance, Software Assistance, Upper Limb Assisted Exercise, Lower Limb Wearable Device, and Placebo. The size of each node reflects the total number of participants involved in that intervention across all included studies. The width of each connecting line corresponds to the number of randomized controlled trials (RCTs) directly comparing the two connected interventions. Thicker lines indicate more available direct comparisons.

**Figure 4 healthcare-13-02999-f004:**
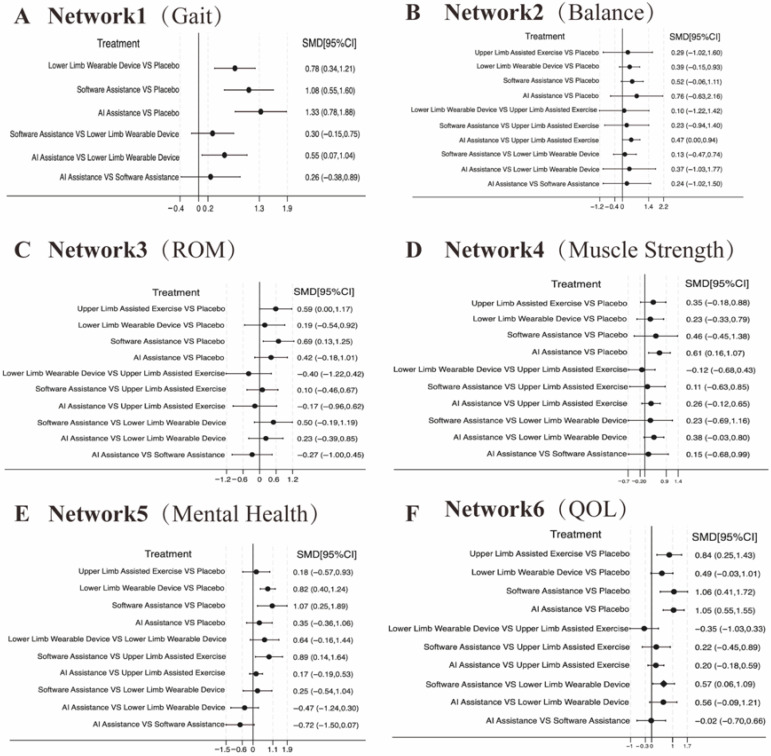
Forest plots of network meta-analyses comparing the effects of six intervention modalities (AI assistance, software assistance, upper limb assisted exercise, lower limb wearable device, placebo) across six key outcomes in older adults: (**A**) Gait Performance, (**B**) Balance Ability, (**C**) Joint Range of Motion (ROM), (**D**) Muscle Strength, (**E**) Mental Health, and (**F**) Quality of Life (QOL). Effect sizes are expressed as standardized mean differences (SMDs) with 95% confidence intervals (CIs).

**Figure 5 healthcare-13-02999-f005:**
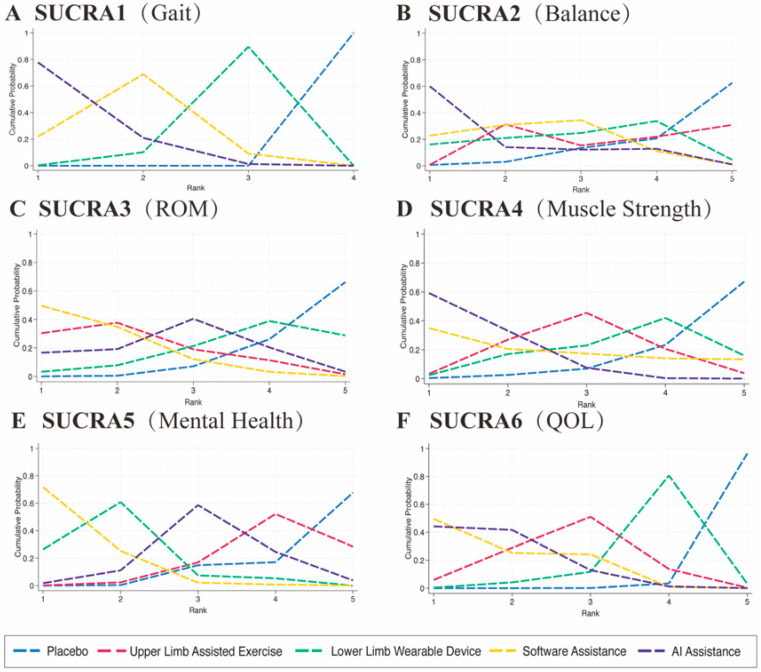
SUCRA-based cumulative ranking curves for six outcome indicators. Each panel presents the cumulative probability curves of the compared interventions for a specific outcome: (**A**) Gait Performance, (**B**) Balance Ability, (**C**) Joint Range of Motion (ROM), (**D**) Muscle Strength, (**E**) Mental Health, and (**F**) Quality of Life (QOL).The SUCRA method ranks the treatments from the most to the least effective based on probability distributions. A curve closer to the top-left corner indicates a higher probability of achieving better rankings.

## Data Availability

No new data were created or analyzed in this study. Data sharing is not applicable to this article.

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
