# Peer review of "Comparing the Effects of AI-Assisted and Traditional Exercise on Physical Health Outcomes in Older Adults: A Systematic Review and Meta-Analysis"

_healthcare, 2025, doi:10.3390/healthcare13232999_

Round 1
Reviewer 1 Report
Comments and Suggestions for Authors
The article offers a valuable synthesis of current evidence and highlights promising directions in the use of artificial intelligence for rehabilitation in the elderly, and provides a better perspective for the increasingly large ageing population.
The content is clear and well-structured. My only suggestion is that the distinctions between AI-assisted and software-assisted interventions could be more sharply defined.
About the form, I have some comments.
In the text body, reference numbers should be placed in square brackets [ ].
There are some minor editing problems with the spaces before parentheses.
Figure 3 is difficult to read. Perhaps changing the 6 network diagrams from 2 into 3 rows will make the adjacent text more legible.
Figure 6 deserves the source mention, in case it will be used in future by other authors.
In the Statement of Ethics, the phrase This tool assisted in ensuring that the manuscript's statements were presented clearly and understandably, facilitating better communication of the research findings. is three times repeated.
In the Data Availability section, which I consider not relevant for this article type, the Competing Interests phrase is repeated.
Author Response
Content:
Point 1: The distinction between AI-assisted and software-assisted interventions is not sharply defined.
Response1: We believe that the distinction between AI-assisted and software-assisted interventions is clearly defined. In this study, the classification was based on the rationale summarized in Supplementary Table S5: Rationale for Intervention Classification of Included Studies. Specifically, interventions were defined as AI-assisted when using machine learning or deep learning algorithms to achieve real-time recognition, adaptive adjustment, or predictive feedback, and as software-assisted when using technological media for training feedback without algorithmic adaptability or learning functions. This operational distinction ensures internal consistency across studies with heterogeneous designs and outcome measures, as detailed in Supplementary Table S5.
Form:
Point 2: In the text body, reference numbers should be placed in square brackets [ ].
Response 2: Thank you for pointing this out. We have revised the manuscript accordingly, and now all in-text citations are consistently presented in square brackets [ ].
Point 3: There are some minor editing problems with the spaces before parentheses.
Response 3: We have carefully checked the entire manuscript to ensure that all parentheses are properly formatted with appropriate spacing and that only English-style brackets are used throughout the text.
Point 4: Figure 3 is difficult to read; the six network diagrams could be reorganized into three rows.
Response 4: Thank you for your helpful suggestion. We have improved the readability of Figure 3 by reorganizing the six network diagrams into three horizontal rows, ensuring a clearer visual layout and better alignment with the adjacent text. We believe this adjustment enhances both the clarity and interpretability of the figure.
Point 5: Figure 6 deserves the source mention in case it is used by future authors.
Response 5: Figure 6 was created using BioRender, and we have clearly indicated this information in the figure legend as follows: “Source: Created by the authors using BioRender (www.biorender.com).” We hope this clarification adequately addresses your concern.
Point 6: In the Statement of Ethics, the phrase “This tool assisted in ensuring that the manuscript’s statements were presented clearly and understandably, facilitating better communication of the research findings.” is repeated three times.
Response 6: Thank you for your careful observation. We have revised the manuscript to remove the repeated phrase and have placed the Statement of Ethics section at the end of the manuscript for better clarity and structure. In the latest version, we have removed all duplicate content to ensure that this issue will not occur again.
Point 7: In the Data Availability section, which is not relevant for this article type, the Competing Interests phrase is repeated.
Response 7: We have revised the manuscript to remove the repeated Competing Interests phrase from the Data Availability section. This section has now been corrected and streamlined to match the requirements of this article type. In the latest version, the redundancy has been fully removed to ensure accuracy and clarity.
Reviewer 2 Report
Comments and Suggestions for Authors
General Comment
Thank you for the opportunity to review your manuscript. This study is a systematic review and meta-analysis examining the effects of exercise using artificial intelligence. However, due to the following fundamental issues, I recommend rejecting the manuscript in its current form.
Comment #1
While the keyword “AI” is used, employing the OR operator makes it easier to find specific exercises like “training,” “Ai chi,” and “T'ai chi.” Since abbreviations are typically used after showing the full spelling, it might be better to avoid the search term “AI.”
Comment #2
The definition of “AI” in this study is ambiguous regarding whether it refers to exercise instruction using robots or the use of artificial intelligence to create optimal exercise programs. The criteria for selecting the literature should be clarified.
Comment #3
The list of literature used in the analysis was not provided, and the “Data Availability” section contains the same information as the “Declaration of Competing Interests,” making it impossible to assess the validity of the analysis results during the peer review stage. For example, could you make the references used in this study available for readers to verify, such as the literature cited in "https://www.thelancet.com/journals/landia/article/PIIS2213-8587(21)00051-6/fulltext?s=09," perhaps as supplementary material?
Comment #4
Upon verifying the references adopted in this study, there were several publications , such as “Housman et al. (2008)” and “Maestro et al. (2012)”, that I could not confirm. Furthermore, I found and verified a publication corresponding to “Timmermans et al. (2014),“ but the text does not contain any term corresponding to ”artificial intelligence (AI)." Please re-confirm the appropriateness of the collected literature through verifying the criteria for literature inclusion pointed out in Comment #2 and creating the literature list mentioned in #3.
Minor Comment
Literature Search Strategy
Scopus was not included in the database used.
Author Response
#For reviewer 2
Point 1: While the keyword “AI” was used, employing the OR operator may have unintentionally captured unrelated terms such as “Ai Chi” or “T’ai Chi.” It may be better to avoid the search term “AI.”
Response 1: We appreciate your valuable suggestion. We have revised the manuscript in lines 137–145 to improve the clarity of the search strategy. In particular, in lines 143–144, we explicitly stated that the isolated keyword “AI” was not used, in order to avoid unintentionally capturing unrelated terms (e.g., “Ai Chi,” “T’ai Chi”). The corresponding detailed search strategies have been included in the Supplementary Materials.
Point 2: The definition of “AI” is ambiguous. It is unclear whether it refers to robotic exercise instruction or the use of artificial intelligence to create optimal programs. The inclusion criteria should be clarified.
Response 2:
We appreciate the reviewer’s insightful comment regarding the definition of “AI.” In this study, AI-assisted interventions were clearly defined as those using machine learning or deep learning algorithms to achieve real-time recognition, adaptive adjustment, or predictive feedback, rather than merely robotic exercise instruction. Interventions involving robotic devices were classified as AI-assisted only when they incorporated algorithmic adaptability or intelligent control; otherwise, they were categorized as software-assisted. The detailed rationale for this classification and the inclusion criteria are provided in Supplementary Table S5: Rationale for Intervention Classification of Included Studies. This operational definition ensures a consistent and transparent distinction between AI-assisted and software-assisted interventions across all included studies.
The complete search strategies for each database have been added in the Supplementary Materials (Table S1). In the Methods – Inclusion and Exclusion Criteria section (lines 155–158), we clarified that only rehabilitation-focused exercise programs were included, while general fitness or preventive programs were excluded.In the Introduction (lines 105–110), we explicitly distinguished between AI-assisted and software-assisted interventions to clarify the categorization of multimodal interventions.In the Methods section (lines 180–189), we specified that two independent reviewers assessed study quality using the Cochrane Risk of Bias 2.0 tool, with discrepancies resolved by a third reviewer.
Point 3: The list of literature used in the analysis was not provided, and the Data Availability section repeats the Competing Interests statement. Without a list, it is difficult to validate the results.
Response 3:
Thank you very much for this helpful comment. We have revised the manuscript to address this issue in two ways:
List of included studies: We have included all supplementary materials to address this concern. S1.4 Group Data contains the source data for our meta-analysis, S2. Full Text Articles Included provides the full list of original studies included in the analysis, and S3. Data | AI vs. Traditional NMA presents all meta-analytic results. We believe these materials will help enhance the transparency and validity of our findings.
Data Availability statement: We have corrected the error in this section, which previously repeated the Competing Interests statement. In lines 529–531, we have revised the statement accordingly.
Point 4: Several references (e.g., Housman et al. 2008; Maestro et al. 2012) could not be confirmed. Timmermans et al. (2014) does not contain explicit AI-related terminology. The appropriateness of included studies should be re-verified.
Response 4: We sincerely appreciate this valuable comment. All references included in our analysis have been listed in S2. Full Text Articles Included, which also contains Housman et al. (2008) and Maestro et al. (2012). Regarding Timmermans et al. (2014), we did not classify this study as part of the AI-assisted group; rather, it was categorized under robot-assisted interventions, as the study involved task-oriented upper limb training supported by the Haptic Master robotic device, which provided guided assistance for arm movement during the exercises. We are grateful for your careful review and have further clarified this classification in the revised manuscript to ensure consistency and accuracy.
Point 5: Scopus was not included in the database search.
Response 5: We would like to clarify that Scopus was also searched as part of our database strategy; however, this search did not yield any additional eligible studies beyond those already identified in PubMed, Embase, Web of Science, and the Cochrane Library. To avoid ambiguity, we have revised the manuscript in lines 137–138 to explicitly state that Scopus was included in the database search.
Reviewer 3 Report
Comments and Suggestions for Authors
Dear authors,
First of all, congratulations on this valuable and timely work. The systematic approach and the application of network meta-analysis to compare AI-assisted, software-assisted, and traditional rehabilitation interventions for older adults represent a significant contribution to the field.
However, several aspects could be improved to enhance the clarity and interpretability of the manuscript:
- Introduction: The role of artificial intelligence in the interventions is not clearly contextualized. It would be helpful to clarify whether AI is primarily used for training program design (e.g., exercise prescription, personalization algorithms) or for monitoring and assistance (e.g., feedback systems, sensors, gamification). This distinction would help readers better understand the technological scope of the interventions.
- Methods: The inclusion criteria do not explicitly state whether non-rehabilitation exercise programs (e.g., general fitness or preventive programs) were excluded. If only rehabilitation-focused studies were included, this should be clearly stated, and the title should reflect that scope accordingly.
- Results: While the statistical synthesis is strong, the manuscript lacks essential information on the characteristics of the interventions included in the analysis. It is unclear what types of exercises were used (e.g., strength, coordination), what technologies or platforms were employed, or how intensity, frequency, and adherence were handled. Including a summary table categorizing these features would improve the study’s transparency and real-world applicability. Additionally, it would be helpful to clarify whether the authors had sufficient data to distinguish between AI used for assisting ongoing training sessions versus designing personalized programs.
- Structure: Consider moving the PRISMA diagram and risk of bias results into the Results section, rather than the Methods. This would align more closely with standard reporting practices and improve readability.
Addressing these points would significantly strengthen the clarity, reproducibility, and impact of the manuscript.
Author Response
Point 1: In the Introduction, the role of artificial intelligence in the interventions is not clearly contextualized. It should be clarified whether AI is primarily used for training program design (e.g., exercise prescription, personalization algorithms) or for monitoring and assistance (e.g., feedback systems, sensors, gamification).
Response 1: We sincerely appreciate your valuable comment. We have revised the text to clearly contextualize the role of AI in the included interventions(from 105-110),. In our study, AI was primarily applied for monitoring and assistance, including posture recognition, gait analysis, scenario detection, and real-time feedback. Additionally, some interventions employed AI for personalization algorithms and adaptive training design, such as adjusting exercise prescriptions based on user performance. We emphasized that the core contribution of AI lies in its data-driven monitoring and feedback mechanisms, which enhance adherence, accuracy, and safety of training for older adults.
Point 2: In the Methods, the inclusion criteria do not explicitly state whether non-rehabilitation exercise programs (e.g., general fitness or preventive programs) were excluded. If only rehabilitation-focused studies were included, this should be clearly stated, and the title should reflect that scope accordingly.
Response 2:
Thank you very much for this insightful and valuable comment. In the process of literature search and screening, we found that the vast majority of studies applying AI-assisted exercise interventions for older adults were conducted in rehabilitation or clinical settings, primarily targeting individuals with cognitive decline, Alzheimer’s disease, stroke, or other age-related disorders. In contrast, studies involving healthy older adults or general fitness and preventive programs were extremely limited and often lacked the use of rehabilitation-related equipment or AI-assisted adaptive systems.For this reason, we explicitly restricted our inclusion criteria to rehabilitation-focused programs, which aligns with the actual current application trend of AI in geriatric health — namely, that AI-assisted technologies are most frequently used for rehabilitation, motor recovery, and cognitive enhancement among older adults with clinical conditions.
Therefore, general fitness or preventive interventions were excluded from this review, and this clarification has been added to the Methods – Inclusion and Exclusion Criteria section (lines 155–158). We believe this revision makes our study scope clearer and more consistent with real-world practice in the field.
Point 3: In the Results, the manuscript lacks details on the characteristics of the interventions included (e.g., exercise type, technologies/platforms, frequency, intensity, adherence). A summary table categorizing these features would improve transparency and real-world applicability. It would also be helpful to clarify whether sufficient data were available to distinguish between AI used for assisting training versus designing personalized programs.
Response 3: Thank you very much for your constructive suggestion. We have now added a detailed summary of the intervention characteristics, including exercise type, technologies/platforms, frequency, intensity, and adherence, in the newly created Supplementary File S4. Detailed Description of Intervention Characteristics. The distinctions between AI used for assisting training and AI for personalized program design are also clarified in this file. All relevant details can be found in Supplementary File S4.
Point 4: Regarding Structure: The PRISMA diagram and risk of bias results are currently located in the Methods section, but should be moved to the Results section for better alignment with reporting standards and readability.
Response 4: Thank you for this valuable suggestion. We have revised the manuscript structure accordingly. The PRISMA flow diagram and the results of the risk of bias assessment have now been moved from the Methods section to the Results section, in line with PRISMA-NMA reporting standards. In addition, we have added a brief explanatory sentence at the end of the corresponding paragraph in the Results section, noting that the risk of bias assessment is presented in Figure 2. We believe these adjustments improve both the readability and compliance of the manuscript.
Reviewer 4 Report
Comments and Suggestions for Authors
This study evaluates the effectiveness of AI-assisted, software-assisted, and conventional rehabilitation approaches in older adults using a network meta-analysis (NMA). The review is relevant; however, it requires substantial improvements strengthen its scientific contribution.
- Title and abstract are appropriate
- While the comparative evaluation of AI-assisted interventions is valuable, similar meta-analyses in rehabilitation technology have recently been published. The manuscript must highlight what new insights this study contributes beyond prior work.
- The integration of both physical and psychological outcomes is a strength, but the novelty of combining AI with traditional rehabilitation models should be further emphasized.
- The search strategy and inclusion/exclusion criteria must be described in greater detail, including trial quality assessments and how multimodal interventions were categorized.
- Reporting of statistical methods (SUCRA, heterogeneity, sensitivity analyses) is brief. More transparency on how inconsistency across the network was addressed is required.
- The conclusion overstates the superiority of AI-assisted programs despite the subgroup finding that “differences among novel interventions were not statistically significant.” This discrepancy needs to be carefully reconciled.
- Clinical implications are only broadly stated. The authors should provide specific recommendations for real-world adoption and limitations regarding cost, accessibility, and patient adherence.
- Clear figures (network plots, forest plots, SUCRA rankings) are essential and should be presented.
- References should be updated to include the latest meta-analyses (2023–2025) in AI-driven rehabilitation and digital health interventions, to position the contribution more clearly.
- Here are few recent studies which may add value to the review. It can be include in preface to strengthen the case of review and better understanding of its need. https://doi.org/3389/fpubh.2025.1635475, , https://doi.org/10.1186/s12984-024-01497-5, https://doi.org/10.1186/s12984-024-01369-y, https://doi.org/10.3389/fnagi.2024.1327397
Author Response
Point 1: While the comparative evaluation of AI-assisted interventions is valuable, similar meta-analyses in rehabilitation technology have recently been published. The manuscript must highlight what new insights this study contributes beyond prior work.
Response 1: Thank you for your insightful comment. In the revised manuscript (Lines 110–116), we have clearly emphasized the novel contributions of this study compared with previous meta-analyses. Specifically, earlier research has mainly focused on single-point or device-level applications (e.g., exoskeleton systems, neuromuscular control, or localized AI feedback). In contrast, our study is, to our knowledge, the first to conduct a network meta-analysis comparing AI-assisted, software-assisted, and traditional rehabilitation interventions across six multidimensional outcome domains (gait, balance, range of motion, muscle strength, mental health, and quality of life). This comprehensive comparison provides an integrated understanding of how AI-based rehabilitation performs relative to both conventional and software-assisted approaches, offering new evidence for clinical and community-level implementation.
Point 2: The integration of both physical and psychological outcomes is a strength, but the novelty of combining AI with traditional rehabilitation models should be further emphasized.
Response 2:
In the revised manuscript, we have explicitly highlighted the novel contributions of this study and clarified how it differs from prior research (Lines 110–116 and 452–457). Previous meta-analyses in rehabilitation technology have mainly focused on single-point or device-level AI applications (e.g., exoskeletons, neuromuscular control, or isolated feedback systems), whereas our study is, to our knowledge, the first to perform a network meta-analysis comparing AI-assisted, software-assisted, and traditional rehabilitation interventions across six outcome domains — gait, balance, range of motion (ROM), muscle strength, mental health, and quality of life (QoL).
Furthermore, we emphasized the innovative integration of AI with traditional rehabilitation models, highlighting how this hybrid “AI + traditional rehabilitation” framework combines data-driven adaptive feedback and therapist-guided personalization. This approach not only bridges physical and psychological outcomes but also provides a scalable, human-centered rehabilitation model that enhances both engagement and accessibility for older adults.
Point 3: The search strategy and inclusion/exclusion criteria must be described in greater detail, including trial quality assessments and how multimodal interventions were categorized.
Response 3:
We have revised the manuscript to provide greater detail regarding the search strategy, inclusion/exclusion criteria, trial quality assessment, and categorization of multimodal interventions:
Search strategy: The complete search strategies for each database have been added in the Supplementary Materials (S1).
Inclusion/exclusion criteria: In lines 155–158, we clarified that only rehabilitation-focused exercise programs were included, while general fitness or preventive programs were excluded.
Categorization of multimodal interventions: We explicitly distinguished between AI-assisted and software-assisted interventions in lines 105–110, thereby clarifying how multimodal interventions were categorized.We categorized all included studies into four types of interventions based on their technological features and training focus, as detailed in Supplementary Table S5. AI-assisted interventions referred to those incorporating artificial intelligence algorithms such as deep learning, CNN, LSTM, posture or gait recognition, or real-time adaptive feedback, enabling data-driven personalization and predictive control. Software-assisted interventions involved VR systems, wearable sensors, or gamified rehabilitation platforms that provided interactive feedback but lacked self-learning algorithms. Upper-limb rehabilitation interventions focused on robotic or task-oriented training for hand and arm functions, while lower-limb rehabilitation interventions targeted gait, balance, or walking training using exoskeletons or mechanical support systems without AI components. This classification ensured internal consistency across heterogeneous study designs and allowed for a comprehensive comparison of different rehabilitation modalities in older adults.
Trial quality assessment: We detailed in lines 180–189 that two independent reviewers assessed study quality using the Cochrane Risk of Bias 2.0 tool (RoB 2.0), with discrepancies resolved by a third reviewer. The specific responsibilities of each author are further described in the Author Contributions section.
We hope these revisions enhance the transparency of the methodology and adequately address your concerns.
Point 4: Reporting of statistical methods (SUCRA, heterogeneity, sensitivity analyses) is brief. More transparency on how inconsistency across the network was addressed is required.
Response 4: Thank you very much for your insightful suggestion. We have revised the Results section to enhance the transparency and completeness of our statistical reporting. Specifically, additional descriptions of SUCRA stability, heterogeneity, and network inconsistency testing have been incorporated for all six outcome domains. The revisions are as follows: lines 242–245 (Gait Performance), 265–269 (Balance Ability), 288–291 (Range of Motion), 309–312 (Muscle Strength), 331–334 (Cognitive Function), and 353–356 (Health-Related Quality of Life). These updates clarify how inconsistency was examined using both global and local approaches (e.g., node-splitting method, P > 0.05 indicating good consistency), and confirm the robustness of SUCRA rankings through sensitivity analyses. These improvements ensure that the statistical methodology is more transparent, replicable, and aligned with current standards for network meta-analysis reporting.
Point 5: The conclusion overstates the superiority of AI-assisted programs, despite subgroup findings that “differences among novel interventions were not statistically significant.” This discrepancy needs to be reconciled.
Response 5: Thank you very much for this valuable comment. We have revised the Conclusion section (lines 506–510) to provide a more balanced interpretation of the findings. The updated text clarifies that although all active interventions outperformed conventional care, the advantages of AI-assisted interventions were promising but not statistically superior compared with other novel approaches. We also emphasize that AI serves as a complementary and scalable tool for enhancing rehabilitation precision rather than an absolutely superior method.
Point 6: Clinical implications are only broadly stated. Specific recommendations should be provided regarding real-world adoption and limitations (cost, accessibility, adherence).
Response 6: We appreciate your valuable suggestion regarding the need for more specific clinical implications and limitations. In the revised version, we have expanded the Discussion section (Lines 452–457 and 467–469) to provide clear recommendations for real-world adoption. These additions include practical considerations such as cost, accessibility, and adherence, along with strategies like simplified user interfaces, government-subsidized equipment, and community-level training programs to enhance equity and sustainability. Furthermore, we emphasized ethical governance and data transparency to ensure safe and responsible implementation of AI-assisted rehabilitation in clinical and community settings.
Point 7: Clear figures (network plots, forest plots, SUCRA rankings) are essential and should be presented.
Response 7: Thank you for your valuable suggestion. We have carefully checked all the figures and ensured that the network plots, forest plots, and SUCRA rankings are clearly presented with high resolution and consistent formatting. All figures have been exported in TIFF format to ensure maximum clarity and print quality. We believe these revisions enhance the readability and visual precision of the manuscript.
Point 8: References should be updated to include the latest meta-analyses (2023–2025) in AI-driven rehabilitation and digital health interventions.
Response 8:
Thank you very much for this valuable suggestion. We have updated the reference list by replacing the earlier citations with three of the four reviewer-recommended meta-analyses. These newly added studies strengthen the relevance and timeliness of our literature base.
In the revised manuscript, we also explicitly emphasized how our work differs from these previous studies(Line110-116). Specifically, prior research has primarily focused on single-point technologies or device-level applications (e.g., exoskeleton systems, neuromuscular control, or localized AI feedback), whereas comparative evidence across different types of AI interventions and multidimensional health outcomes has been lacking. Our study is, to our knowledge, the first to perform a network meta-analysis comparing AI-assisted, software-assisted, and traditional rehabilitation interventions across six outcome domains — gait, balance, range of motion (ROM), muscle strength, mental health, and quality of life (QoL).
One of the four suggested references could not be retrieved because the DOI provided did not correspond to an existing record. We have attached a screenshot of this reference and link for transparency in below materials. https://doi.org/3389/fpubh.2025.1635475
Round 2
Reviewer 2 Report
Comments and Suggestions for Authors
Thank you for revisions and supplying additional materials. I have confirmed that the changes made in response to my comments are generally appropriate. Regarding the revised content, I recommend the following additional revisions.
- The description of the search engine used in the abstract does not include "Scoups."
- The first letter of “houseman” in Figure 1 should be upper case.
Author Response
Comments 1:
The description of the search engine used in the abstract does not include “Scoups.”
Response 1:
Thank you very much for your careful review and for pointing out this omission. We have revised the abstract to include “Scopus” in the list of search engines used.This revision can be found in the Abstract (Page 1, Line 7) of the revised manuscript.
Comments 2:
The first letter of “houseman” in Figure 1 should be upper case.
Response 2:
Accordingly, we have corrected “houseman” to “Houseman” in Figure 2.This revision can be found in Figure 2 (Page 6) of the revised manuscript.
Reviewer 3 Report
Comments and Suggestions for Authors
I would like to sincerely thank the authors for the considerable efforts made in revising the manuscript. The updated version reflects significant improvements in both the structure and clarity of the content. The incorporation of the reviewers’ suggestions has notably enhanced the methodological transparency and the interpretability of the results, particularly regarding the comparative effectiveness of AI-assisted and traditional exercise interventions.
Congratulations on this comprehensive and well-conducted systematic review and network meta-analysis. The manuscript addresses a timely and relevant topic in geriatric rehabilitation.
Author Response
Thank you very much for your kind and encouraging comments. We sincerely appreciate your recognition of our efforts in revising the manuscript. Your constructive feedback and thoughtful guidance throughout the review process have been invaluable in improving the quality, clarity, and rigor of our work.
We are truly grateful for your positive evaluation and are pleased that the revised manuscript meets the expectations of the reviewers and editors. It has been our honor to contribute to the discussion on AI-assisted and traditional exercise interventions in geriatric rehabilitation.
Reviewer 4 Report
Comments and Suggestions for Authors
No more comments. All queries resolved.
Check the ref 10, looks incomplete. Correct in proof.
Author Response
Comments1:Check the ref 10, looks incomplete. Correct in proof.
Response1:Thank you for pointing this out. We have checked Reference 10 and updated it to the full citation in the manuscript.